# Comparative Analysis of the Kinetic Behavior of Systemic Inflammatory Markers in Patients with Depressed versus Preserved Left Ventricular Function Undergoing Transcatheter Aortic Valve Implantation

**DOI:** 10.3390/jcm10184148

**Published:** 2021-09-15

**Authors:** Haitham Abu Khadija, Gera Gandelman, Omar Ayyad, Lion Poles, Michael Jonas, Offir Paz, Sorel Goland, Sara Shimoni, Valery Meledin, Jacob George, Alex Blatt

**Affiliations:** Kaplan Heart Center, Kaplan Medical Center, Rehovot, Affiliated with the Hebrew University, Jerusalem 76610, Israel; haitham2048@yahoo.com (H.A.K.); Gera_G@clalit.org.il (G.G.); omeray@clalit.org.il (O.A.); Lion_P@clalit.org.il (L.P.); MichaelYo2@clalit.org.il (M.J.); ofirrpa@clalit.org.il (O.P.); Sorel_G@clalit.org.il (S.G.); Sarah_S2@clalit.org.il (S.S.); Valeri_M@clalit.org.il (V.M.); Kobige@clalit.org.il (J.G.)

**Keywords:** TAVI, TAVR, inflammation, neutrophil–lymphocyte ratio, NLR, LVEF, reduced LVEF

## Abstract

Background: Prior studies have proven the safety and efficacy of transcatheter aortic valve implantation (TAVI) in patients with reduced left ventricular (LV) function. This study’s aim was to investigate periprocedural inflammatory responses after TAVI. Methods: Patients with severe symptomatic aortic stenosis and reduced LV function who underwent transfemoral TAVI were enrolled. A paired-matched analysis (1:2 ratio) was performed using patients with preserved LV function. Whole white blood cells (WBC) and subpopulation dynamics as well as the neutrophil to lymphocyte ratio (NLR) were evaluated at different times. Results: A total of 156 patients were enrolled, including 52 patients with LVEF < 40% 35.00 [30.00, 39.25] and 104 with LVEF > 50% 55.00 [53.75, 60.0], *p* < 0.001. Baseline NLR in the reduced LV function group was significantly higher compared to the preserved LV function group, 2.85 [2.07, 4.78] vs. 3.90 [2.67, 5.26], *p* < 0.04. After a six-month follow-up, the inflammatory profile was found to be similar in the two groups, NLR 2.94 [2.01, 388] vs. 3.30 [2.06, 5.35], *p* = 0.288. No significant mortality differences between the two groups were observed in the long-term outcome. Conclusions: TAVI for severe symptomatic aortic stenosis, with reduced LV function, was associated with an improvement in the inflammatory profile that may account for some of the observable benefits of the procedure in this subset of patients.

## 1. Introduction

Surgical aortic valve replacement performed in patients with reduced left ventricular ejection fraction (LVEF) is associated with worse patient outcomes [1,2]. However, transcatheter aortic valve implantation (TAVI) does not increase mortality or heart failure readmissions [3,4]. Moreover, patients with reduced LV function obtain better outcomes from TAVI compared to patients with preserved LV function [5,6,7,8].

Atherosclerosis and degenerative aortic stenosis have common risk factors and etiopathogenesis, and inflammation appears to play an important role in this respect. Furthermore, systemic inflammation has been recognized as a dominant feature in the progression of heart failure [9,10,11]. Specifically, inflammation has been linked to disease development, progression, and associated complications, and is predictive of poor patient outcomes, independent of LVEF [12,13,14]. One such inflammatory biomarker is the NLR, which is derived from routine complete blood counts. The NLR leverages early hematological observations in which the absolute neutrophil count is positively associated with cardiovascular events, while the absolute lymphocyte count is negatively associated with cardiovascular events [15,16]. Moreover, while total white blood cell counts have been shown to predict cardiovascular risk, the NLR appears to be a superior predictor [17]. Some studies have evaluated the NLR as a predictor of total mortality in the setting of acute coronary interventions [17,18] and heart failure [19].

We aimed to compare the kinetic behavior of inflammatory profile after TAVI in patients with preserved or reduced LV function by measuring the NLR biomarker as a potential mediator of outcome improvement from the procedure.

## 2. Methods

### 2.1. Study Population

We retrospectively included patients with severe symptomatic aortic stenosis who underwent TAVI in our center between January 2010 and March 2020. This study was approved by Kaplan Medical Center’s IRB. We excluded patients with chronic systemic inflammatory or autoimmune diseases, acute infections, hematological disorders, malignancies, dialysis, or treatment with agents affecting the white blood cell count, e.g., corticosteroids. Patients who did not have repeated blood tests and those with periprocedural death (up to 72 h after TAVI) were excluded as well.

### 2.2. TAVI Procedure

The type and size of the valves were at the discretion of the local heart team, and a decision was made according to the patient’s anatomical and clinical characteristics. Patients treated with BEV were implanted with either SAPIEN, SAPIEN XT, or S3 (Edwards Lifesciences, Irvine, CA, USA) valves. Patients treated with SEV were implanted with either CoreValve, Evolut PRO, or Evolut R (Medtronic, Inc., Minneapolis, MN, USA) valves. The small number of subjects treated with other valves were also excluded from the analysis. Transfemoral vascular access and closure was performed using the percutaneous approach with the safety wire technique and the Prostar XL (Abbott Vascular, Redwood City, CA, USA) vascular closure device. The procedure duration was calculated as “skin-to-skin”, i.e., time 0 marked the beginning of arterial blood pressure regulation by the accessory support access and the final time was represented by the termination of this accessory support access. Following that, we also used local anesthesia with conscious sedation as a first-line approach. All patients received unfractionated heparin to maintain a minimum active clotting time of >250 s after the insertion of the femoral sheet. Protamine (1 mg for each 100 U of heparin, maximal dose 50 mg) was administered at the time of vascular closure if needed. The use of prophylactic antibiotics during the procedure or hospital stay was routinely avoided. Aspirin was recommended before TAVI. Dual-antiplatelet treatment with 100 mg of aspirin and 75 mg of clopidogrel was started the day before the procedure and followed thereafter for six months, except for patients requiring chronic oral anticoagulation.

### 2.3. Inflammatory Markers

Baseline characteristics, procedural data, and clinical outcomes were collected. Blood samples were obtained using a 21G sterile syringe without stasis. Laboratory analyses were performed before the procedure, during the patient’s postprocedural intensive care unit stay on a daily basis, and at the physician’s discretion in the cardiology ward. These were retrospectively collected. The neutrophil–lymphocyte ratio (NLR) was calculated using this formula: NLR = absolute neutrophils count/absolute lymphocytes count.

### 2.4. Definition Criteria for Events

All the outcome definitions were strictly determined according to the Valve Academic Research Consortium 2 (VARC-2) criteria. All of these standardized endpoint definitions together represent the major adverse cardiovascular events (MACE) in our study. Clinical follow-up included 30-day and six-month visits after hospital discharge. The follow-ups were performed on site.

### 2.5. Matching

The reduced LV function cohort (LVEF < 40%) was compared to a preserved LV function cohort (LVEF > 50%), using matched-pair grouping for statistical analysis in a retrospective and descriptive manner. Patients suffering mid-range LV dysfunction, i.e., LVEF between 40% to 50%, or severely depressed LV function (LVEF ≤ 25%), were excluded. Matched patients were also selected during the same time period out of 104 patients undergoing TAVI. Age ± 5 years, sex, BMI ± 5, dialysis and additive EuroSCORE ± 5 were used for matched-pair analysis (1:2). Thus, a total of 156 patients were matched and compared. We performed a propensity score to build the matching groups.

### 2.6. Statistical Analysis

Data are presented as median values (with ranges in parentheses). Continuous variables between the various study groups were tested for normality with a Shapiro–Wilk test and when an abnormal distribution was found, a Mann–Whitney test was performed. When the distribution was normal, a *t*-test was used. Pearson’s chi-square test was performed for categorical variables when appropriate. Main effect estimates are presented with their 95% confidence interval. Statistical analysis was performed using SPSS software (IBM SPSS Statistics for Windows, Version 25.0., Armonk, NY, USA) and R programming version 4.0.2 (The R Foundation for Statistical Computing, Vienna, Austria; http://www.R-project.org) (accessed date 1 June 2021) for all analyses. Where the *p* values were less than 0.05, these were considered statistically significant.

## 3. Results

### 3.1. Pre-Procedural Data

During the ten-year study period, 370 consecutive patients were enrolled. The study flowchart is shown in Figure 1. According to our definitions, a total of 56 patients were excluded. The analyzed population included 314 patients (43% female, with a median age of 81.00 [76.00–85.00] years old), with severe symptomatic aortic stenosis (transaortic pressure gradient 37.00 [29.50–47.00] mmHg), and high or prohibitive operative risk (an STS score of 8.01 [5.1–10.3]). Baseline and procedural characteristics of the 1:2 matched study population according to ejection fraction are summarized in Table 1.

The clinical profile and executive procedural characteristics showed no significant differences between the two 1:2 matched population group types at baseline. As expected, according to the study aims, compared with patients with preserved LV function, LV function was significantly depressed in the reduced LV group, LVEF 55.00% [53.75, 60.0] vs. 35.00% [30.00, 39.25] (*p* < 0.001). Septum thickness, aortic valve area (AVA), and mean gradient across the aortic valve were significantly higher in the preserved LV function group, 14.00 mm [13.00, 15.0] vs. 12.50 mm [12.00, 14.0] (*p* < 0.001), 0.70 cm * [0.60, 0.80] vs. 0.65 cm * [0.50, 0.70] (*p* < 0.03) and 43.5 mmHg [32.25, 50.75] vs. 32.00 mmHg [27.00, 37.00] (*p* > 0.003). The main statistically significant difference between the two groups was a higher NLR at baseline in the reduced LV function group compared with the preserved LV function group, 2.85 [2.07, 4.78] vs. 3.90 [2.67, 5.26] (*p* < 0.04).

### 3.2. Procedural Data

Inflammatory marker dynamics: baseline total WBC, absolute cell counts of neutrophils, lymphocytes, and NLR and their dynamic changes after TAVI for the total study population are summarized in Table 2. In the entire 1:2 matched population, we noticed that there were significant kinetic changes in the WBC response (*p* value < 0.0001) from admission, to 24 h post procedure, to 72 h post procedure, with significant increases in WBC, neutrophils, and NLR, and significant decreases in absolute lymphocyte counts. Figure 2 shows the dynamic changes of white blood cells and the differential subsets in the entire study population at different times during the first 24 h and 72 h, and at six months post procedure. There was a significant increase in inflammatory markers, including total WBC, neutrophils, and NLR in all the study 1:2 matched population. When comparing the two subgroups as shown in Table 3 and Figure 3, patients with reduced left ventricular function had a significantly more pronounced inflammatory response at baseline. However, six months after TAVI, there was a similar extent of reduction in the inflammatory markers to about the same level of the in-patients, with reduced NLR compared to preserved ejection fraction, NLR 2.94 [2.01, 388] vs. 3.30 [2.06, 5.35], *p* = 0.288. NLR reduction in the low EF group from baseline to 6 months post procedure did not achieve statistical significance, 3.90 [2.67, 5.26] vs. 3.30 [2.06, 5.35] (*p* = 0.32).

**Table 2 jcm-10-04148-t002:** Dynamic changes of WBC and their subpopulations after TAVI of 1:2 matched population.

	Admission	24 h	72 h	6 Months	*p*1	*p*2	*p*3
WBC (K/μL)	7.46 ± 2.26	10.08 ± 3.55	9 ± 2.91	7.47 ± 2.38	<0.0001	<0.0001	1
Absolute neutrophils (K/μL)	4.97 ± 2.06	8.19 ± 3.43	6.73 ± 2.79	4.89 ± 2.04	<0.0001	<0.0001	1
Absolute lymphocytes (K/μL)	1.67 ± 1.1	1.1 ± 0.76	1.29 ± 0.59	1.76 ± 1.09	<0.0001	<0.0001	1
NLR	3.72 ± 2.8	9.76 ± 7.29	6.52 ± 4.66	3.36 ± 2.23	<0.0001	<0.0001	1

Values are mean ± SD; *p*1 = Comparison of pre-procedural values with those at 24 h; *p*2 = Comparison of pre-procedural values with those at 72 h; *p*3 = Comparison of pre-procedural values with those at 6 months.

**Table 3 jcm-10-04148-t003:** Timeline of dynamic changes of WBCs and its components of 1:2 matched population.

	Overall	EF ≥ 50	EF ≤ 40	*p* Value
	*N* = 156	*N* = 104	*N* = 52	
Admission				
WBC (K/μL)	7.10 [5.89, 8.43]	7.06 [5.90, 8.14]	7.25 [5.68, 8.90]	0.565
Absolute neutrophils (K/μL)	4.70 [3.51, 6.11]	4.59 [3.52, 5.93]	4.88 [3.48, 6.60]	0.316
Absolute lymphocytes (K/μL)	1.31 [1.10, 1.90]	1.40 [1.14, 1.90]	1.28 [1.00, 1.78]	0.215
NLR	3.21 [2.22, 5.00]	2.85 [2.07, 4.78]	3.90 [2.67, 5.26]	0.04
24 h post-procedure				
WBC (K/μL)	9.53 [7.40, 11.80]	9.65 [7.57, 11.88]	9.10 [6.88, 11.10]	0.371
Absolute neutrophils (K/μL)	7.65 [5.50, 9.46]	7.70 [5.50, 9.58]	7.60 [4.97, 9.40]	0.434
Absolute lymphocytes (K/μL)	0.95 [0.66, 1.30]	1.00 [0.70, 1.30]	0.90 [0.60, 1.20]	0.421
NLR	7.83 [5.48, 12.07]	7.40 [5.48, 12.18]	8.35 [5.48, 10.47]	0.678
72 h post-procedure				
WBC (K/μL)	8.49 [7.10, 10.38]	8.55 [7.10, 10.40]	8.30 [7.10, 9.70]	0.497
Absolute neutrophils (K/μL)	6.40 [4.83, 8.16]	6.52 [4.88, 8.22]	6.00 [4.83, 7.65]	0.436
Absolute lymphocytes (K/μL)	1.10 [0.80, 1.49]	1.10 [0.88, 1.49]	1.00 [0.80, 1.48]	0.468
NLR	5.71 [3.77, 8.09]	5.65 [3.82, 8.15]	6.16 [3.48, 8.01]	0.863
6 months post-procedure				
WBC (K/μL)	7.10 [5.84, 8.65]	7.05 [5.72, 8.24]	7.20 [6.00, 9.30]	0.38
Absolute neutrophils (K/μL)	4.60 [3.40, 5.95]	4.50 [3.40, 5.40]	4.60 [3.55, 6.55]	0.288
Absolute lymphocytes (K/μL)	1.60 [1.15, 1.92]	1.60 [1.17, 2.08]	1.60 [1.20, 1.86]	0.844
NLR	3.00 [2.02, 4.35]	2.94 [2.01, 3.88]	3.30 [2.06, 5.35]	0.288

Values are presented as: median (ranges). Abbreviations: WBC = white blood cells; NLR = neutrophils to lymphocytes ratio. In a variable analysis for factors that may influence inflammatory markers post-TAVI for the whole study population, we found significant differences only in the procedure duration and contrast volume injected (Table 4).

**Table 4 jcm-10-04148-t004:** Variable analysis of factors that may influence inflammatory markers post-TAVI for the total population.

		Post-Procedure 24 h—WBC		Post-Procedure 24 h—NEUT Abs		Post-Procedure 24 h—LYMP Abs		NLR 24 h	
		Median (IQR)	*p*-Value	Median (IQR)	*p*-Value	Median (IQR)	*p*-Value	Median (IQR)	*p*-Value
Total		9.53 (7.4–11.8)		7.65 (5.5–9.51)		0.95 (0.62–1.3)		7.83 (5.46–12.14)	
Gender	females	9.6 (7.15–11.41)	0.634	7.77 (5.5–9.4)	0.952	0.9 (0.6–1.3)	0.736	8.02 (5.46–11.5)	0.696
	males	9.4 (7.4–12.4)		7.6 (5.5–10.1)		1 (0.7–1.3)		7.54 (5.5–12.36)	
Age *, years		0.053	0.511	−0.001	0.993	−0.042	0.604	0.017	0.829
BMI *, kg/m^2^		0.112	0.165	0.097	0.229	0.161	0.046	−0.039	0.627
HTN	no	8.7 (7.53–13.4)	0.860	6.9 (5.5–12)	0.975	0.8 (0.6–1.2)	0.521	10.46 (5.28–17.25)	0.411
	yes	9.63 (7.15–11.8)		7.685 (5.5–9.455)		0.975 (0.66–1.3)		7.63 (5.48–11.41)	
Diabetes	no	9.67 (7.15–11.87)	0.956	7.67 (5.4–9.67)	0.944	0.94 (0.7–1.3)	0.889	7.66 (5.18–12.14)	0.446
	yes	9.53 (7.6–11.67)		7.625 (6.17–9.2)		0.975 (0.6–1.3)		7.87 (5.87–11.33)	
AF	no	9.4 (7.53–11.87)	0.613	7.7 (5.7–9.6)	0.251	0.9 (0.7–1.3)	0.272	8.13 (5.62–12.77)	0.075
	yes	9.685 (7.03–11)		7 (5.1–9.2)		1.035 (0.7–1.3)		6.91 (4.78–10)	
CAD	no	9.45 (7.515–11.555)	0.994	7 (5.5–9.4)	0.769	0.94 (0.7–1.3)	0.853	7.54 (5.18–12.2)	0.856
	yes	9.63 (7–11.87)		7.7 (5.3–9.51)		1 (0.6–1.3)		7.83 (5.78–10.85)	
PAD	no	9.565 (7.45–12.085)	0.452	7.67 (5.5–9.79)	0.366	0.95 (0.6–1.3)	0.865	7.9 (5.26–12.77)	0.486
	yes	9.415 (6.85–11)		7.425 (5.05–9)		0.935 (0.7–1.25)		7.2 (5.73–9.64)	
Baseline echo									
LVEF (%) *		0.146	0.070	0.148	0.066	0.036	0.656	0.051	0.532
Septum thickness (mm) *		0.132	0.102	0.142	0.079	−0.137	0.091	0.200	0.013
Procedure related									
TAVI types	SEV	9.8 (7.7–12.7)	0.076	7.7 (5.9–10.3)	0.056	1 (0.7–1.3)	0.848	7.85 (5.55–12.16)	0.389
	BEV	9.25 (7.015–10.9)		7.4 (5.11–8.95)		0.975 (0.6–1.3)		7.29 (5.27–10.47)	
Contrast volume (mL) *	0.150		0.062	0.162	0.044	−0.031	0.705	0.188	0.019
Time (minutes) *	0.113		0.165	0.166	0.041	−0.191	0.018	0.275	0.001
Post-dilatation	no	9.615 (7.53–11.8)	0.585	7.7 (5.5–9.4)	0.702	1 (0.7–1.3)	0.450	7.6 (5.46–12)	0.749
	yes	9.15 (7–11.87)		6.92 (5.5–9.6)		0.855 (0.6–1.2)		8.69 (5.5–12.14)	

Data are presented as: median (ranges). Continuous variables between the various study groups were tested for normality by a Shapiro–Wilk test and when abnormal distribution was found, a Mann–Whitney test was performed. When the distribution was normal, a *t*-test was used. * Correlation between two continuous variables were tested by a Pearson’s correlation. Abbreviations: TAVI = transcatheter valve implantation; BMI = Body mass index; AF = atrial fibrillation; CAD = coronary artery disease; PAD = peripheral arterial disease; LVEF = left ventricle ejection fraction; SEV = self-expandable valve; BEV = balloon-expandable valve; WBC = white blood cells; Abs = absolute; NLR = neutrophils to lymphocytes ratio.

### 3.3. Clinical Outcomes

The thirty-day clinical outcomes according to the VARC-2 criteria are summarized in Table 5. We found no differences between the two subgroups according to MACE and mortality. The Kaplan–Meier survival curve at the four-year follow-up is shown in Figure 4. During the ten-year study period, we found no significant differences between the two groups according to the ejection fraction (*p* = 0.35).

## 4. Discussion

We have presented a matched-pair analysis comparing inflammatory biomarker behavior in patients who underwent TAVI for symptomatic severe aortic stenosis in two different cohorts: patients with reduced LV function versus patients with preserved LV function. The major findings of our study were: (i) the enhanced inflammatory state in patients with reduced LV function compared with preserved LV function is reflected by a higher baseline NLR; (ii) the effect of the TAVI on the reduction of the NLR inflammatory marker in the reduced LV group caught up with the NLR in the preserved LV function group after six months of follow-up; (iii) there are similarly favorable clinical outcomes, independent of the baseline LV function.

Inflammation contributes to the pathogenesis and progression of heart failure (HF) [12] as well as degenerative aortic stenosis [20,21]. Elevated inflammatory biomarkers, including NLR, were reported in the TIME-CHF study [9] that enrolled elderly patients with HF and severe aortic stenosis who were candidates for surgical replacement [22,23,24]. Our findings of elevated NLR in patients with severe aortic stenosis and reduced LV function are concordant with the publication of Avci et al. [22], who found higher NLR in patients presenting with these two entities. In our study, we added another parameter when analyzing the effect of TAVI on the inflammatory profile six months after the procedure, which mirrors complete resolution from inflammation-related features associated with the procedure.

Our findings are in agreement with the recent publication of Baratchi et al. [25], who found an anti-inflammatory effect of TAVI reflected by a significant reduction in the NLR. One possible pathophysiologic explanation is that the reduction of NLR after TAVI in patients with reduced LV function may be related to the effect of TAVI in decreasing hemodynamic resistance, which lowers the high shear stress that activates and perpetuates the inflammatory state, as experimentally demonstrated in the Baratchi publication [25].

Other significant baseline characteristic differences found were septum thickness, aortic valve area, and mean aortic valve gradient. These issues reflect the underling pathophysiology of patients with reduced LV function, which are superimposed with significant aortic stenosis activated compensatory pathways (i.e., chamber remodeling). These parameters are not statistically significant in the variable analysis (Table 4).

Finally, as described in the pivotal PARTNER study [5] and other studies [5,6,26], we show that there are no differences in the long-term mortality rates between patients with decreased or preserved LV function after TAVI. The apparent normalization of the inflammatory markers by TAVI, which was more pronounced in the patients with baseline reduced EF, may be associated with the observation that despite the presence of preexisting LV dysfunction, these patients do not have a poorer short-term outcome. However, this should be tested in larger cohorts and for extended periods of follow-up.

To the best of our knowledge, this is the first report that compares the inflammatory state behavior and kinetics in patients undergoing TAVI with low ejection fraction versus those with preserved LV function. In conclusion, we found that TAVI has a more pronounced anti-inflammatory effect in patients with a low baseline ejection fraction, and this effect may partly account for the beneficial clinical response, which is translated to similar outcomes when compared to patients with preserved LV function.

## Figures and Tables

**Figure 1 jcm-10-04148-f001:**
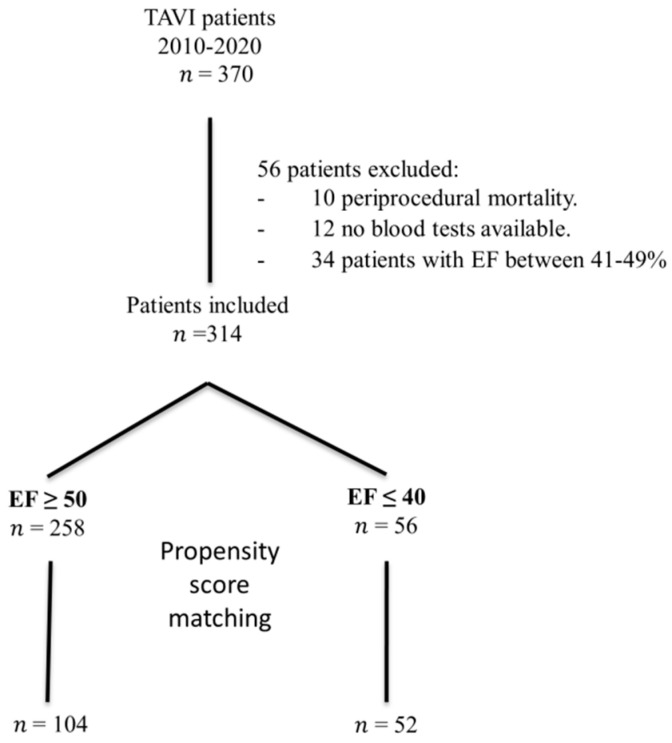
Study flowchart of patients from 2010 to 2019. A total of 370 patients were treated with TAVI. After excluding 56 patients and doing propensity score matching, a total of 156 patients were finally included in the analysis.

**Figure 2 jcm-10-04148-f002:**
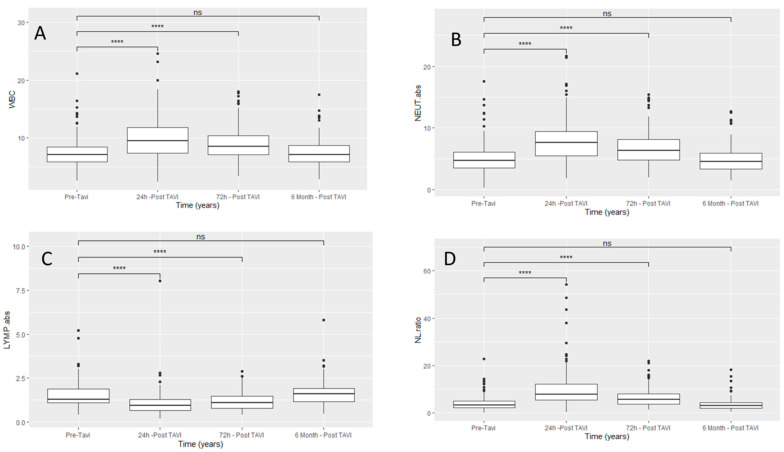
Dynamic changes of leukocytes and its components after TAVI for the matched population, (**A**) dynamic changes in total WBC with time, (**B**) dynamic changes in Neutrophils with time, (**C**) dynamic changes of lymphocytes with time, (**D**) dynamic changes of NL ratio with time. LYMP. abs = Absolute Lymphocytes (K/μL); NEUT. abs = Absolute Neutrophils (K/μL); WBC = White Blood Cells (K/μL); NLR = Neutrophils to Lymphocytes Ratio; ns = not significant; **** = significant (*p* value < 0.01), Dots are the values beyond the extreme line that shows potential outliers.

**Figure 3 jcm-10-04148-f003:**
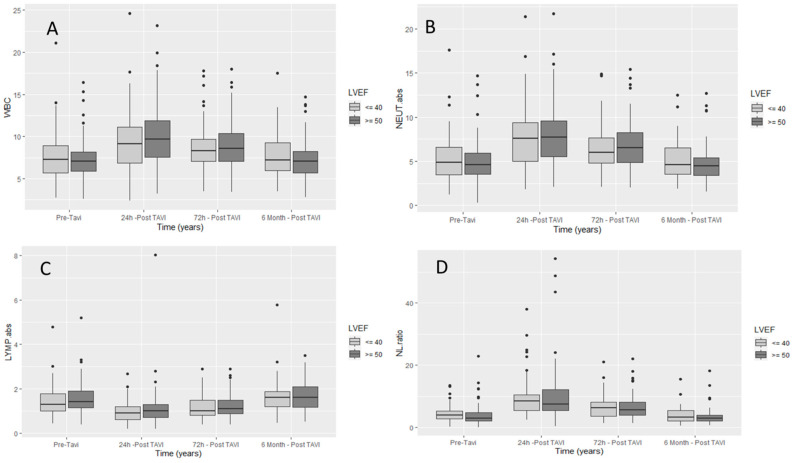
Difference in two matched subgroups (EF ≥ 50 vs. EF ≤ 40) for leukocytes and its components after TAVI, (**A**) difference in total WBC with time, (**B**) difference in Neutrophils with time, (**C**) difference in lymphocytes with time, (**D**) difference in NL ratio with time. LYMP. abs = Absolute Lymphocytes (K/μL); NEUT. abs = Absolute Neutrophils (K/μL); WBC = White Blood Cells (K/μL); NLR = Neutrophils to Lymphocytes Ratio, Dots are the values beyond the extreme line that shows potential outliers.

**Figure 4 jcm-10-04148-f004:**
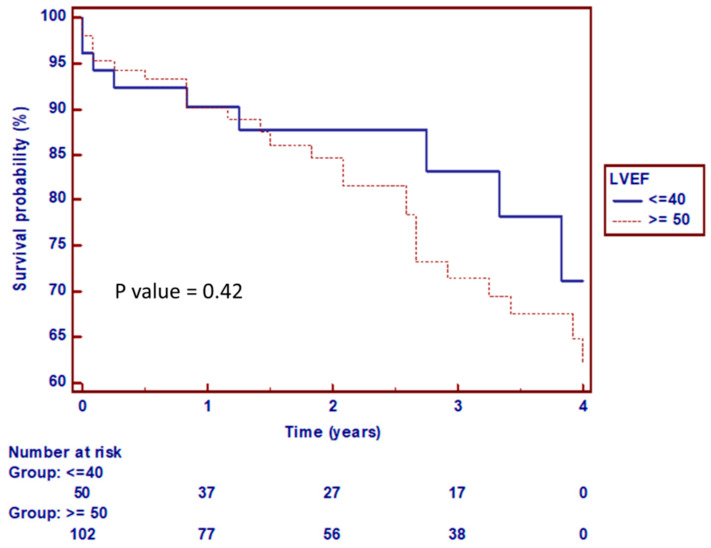
Kaplan Meir curve based on all cause mortality for patients underwent TAVI, showing no difference in survival between the two TAVI matched subgroups according to ejection fraction, LVEF = Left Ventricular Ejection Fraction.

**Table 1 jcm-10-04148-t001:** Baseline characteristics of the 1:2 matched population.

	Overall	EF ≥ 50	EF ≤ 40	*p* Value
Clinical characteristic	*N* = 156	*N* = 104	*N* = 52	
Age, years	81.00 [76.00, 85.00]	81.50 [74.75, 86.25]	81.00 [77.00, 85.00]	0.737
Male (%)	89 (57%)	60 (57.7%)	29 (55.8%)	0.954
Body mass index, kg/m	26.89 [23.44, 29.33]	26.33 [22.72, 30.55]	27.05 [24.08, 29.11]	0.506
Hypertension (%)	145 (93%)	97 (93.3%)	48 (92.3%)	1
Diabetes (%)	67 (43%)	44 (42.3%)	23 (44.2%)	0.954
Dyslipidemia (%)	118 (76%)	75 (72.1%)	43 (82.7%)	0.21
Smoker (%)	21 (13%)	14 (13.5%)	7 (13.5%)	1
Atrial fibrillation (%)	50 (32%)	35 (34.0%)	15 (30.0%)	0.758
Coronary artery disease (%)	71 (46%)	42 (40.8%)	29 (55.8%)	0.11
Peripheral artery disease (%)	32 (21%)	16 (15.4%)	16 (30.8%)	0.042
Previous myocardial infarction (%)	21 (13%)	10 (9.6%)	11 (21.6%)	0.073
Previous stroke (%)	20 (13%)	14 (13.5%)	6 (11.5%)	0.933
Previous pacemaker (%)	22 (14%)	14 (13.5%)	8 (15.4%)	0.935
CABG (%)	12 (0.7%)	5 (7.9%)	7 (17.1%)	0.266
STS Score	8.01 [5.1, 10.3]	8.04 [4.8, 10.2]	8.3 [5.2, 10.6]	0.617
LVEF (%)	52.50 [39.75, 60.00]	55.00 [53.75, 60.0]	35.00 [30.00, 39.25]	<0.001
Septum thickness (mm)	13.00 [12.00, 15.00]	14.00 [13.00, 15.0]	12.50 [12.00, 14.0]	0.001
Aortic valve area (cm)	0.70 [0.60, 0.80]	0.70 [0.60, 0.80]	0.65 [0.50, 0.70]	0.035
Aortic valve gradient- mean (mm Hg)	37.00 [29.50, 47.00]	43.50 [32.25, 50.75]	32.00 [27.00, 37.00]	0.003
Contrast volume (mL)	120.00 [90.00, 157.0]	136.44 (63.55)	122.04 (51.51)	0.158
Time (minutes)	84.00 [68.00, 105.75]	93.44 (33.77)	84.23 (32.07)	0.106
WBC (K/μL)	7.10 [5.89, 8.43]	7.06 [5.90, 8.14]	7.25 [5.68, 8.90]	0.565
Absolute Neutrophils (K/μL)	4.70 [3.51, 6.11]	4.59 [3.52, 5.93]	4.88 [3.48, 6.60]	0.316
Absolute Lymphocytes (K/μL)	1.31 [1.10, 1.90]	1.28 [1.00, 1.78]	1.40 [1.14, 1.90]	0.215
NLR	3.21 [2.22, 5.00]	2.85 [2.07, 4.78]	3.90 [2.67, 5.26]	0.04

Values are presented as: median (ranges) or n (%). Abbreviations: CABG = coronary artery bypass; STS = Society of Thoracic Surgeons; LVEF = left ventricle ejection fraction; WBC = white blood cells; NLR = neutrophils to lymphocytes ratio.

**Table 5 jcm-10-04148-t005:** Thirty-day outcomes of patients after TAVI of 1:2 matched population ^†^.

Variable	Total	EF ≥ 50	EF ≤ 40	*p* Value
Mortality	8 (5.1%)	5 (4.8%)	3 (5.7%)	0.271
Bleeding	8 (0.5%)	6 (5.8%)	2 (3.8%)	0.898
Major vascular complication	13 (0.8%)	9 (8.7%)	4 (7.7%)	1
Stroke	3 (0.1%)	1 (1.0%)	2 (3.8%)	0.536
Myocardial infarction	3 (0.1%)	2 (1.9%)	1 (1.9%)	1
Acute kidney injury	7 (0.5%)	4 (3.8%)	3 (5.8%)	0.891
Arrhythmia	42 (27%)	29 (27.9%)	13 (25.0%)	0.848

† Categorical and nominal variables were reported by prevalence and percentages and were analyzed by Pearson’s chi-square (χ²) test and Fisher’s exact test.

## Data Availability

All relevant data are included within the manuscript. All other data presented in this study are available upon request from the corresponding author.

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
