# Peer review of "Comparative Analysis of the Kinetic Behavior of Systemic Inflammatory Markers in Patients with Depressed versus Preserved Left Ventricular Function Undergoing Transcatheter Aortic Valve Implantation"

_jcm, 2021, doi:10.3390/jcm10184148_

Round 1

Reviewer 1 Report

The authors present a retrospective analysis of patients undergoing TAVR for severe aortic stenosis focusing on kinetic behavior of systemic inflammatory markers, in particular the neutrophil to lymphocyte ratio (NLR) according to systolic LV-function. The authors included 314 patients and performed a 2:1 matching to account for baseline differences between the groups of LVEF<40% and LVEF>50%. They performed serial blood sampling up to 6 months follow-up and further performed a clinical follow-up up to 10 years. The authors found patients with LV<40% to have a higher NRL at baseline compared to patients with preserved LV function. They further found a post-procedural increase of NLR in both groups after TAVR, which normalized at 6 months, showing no significant reduction at 6 months compared to baseline findings.

This analysis has several issues that need to addressed in this Reviewer opinion:

Major comments:

Methods:

  • matching should be described in more detail, which kind of matching? (Propensity-matching is mentioned in Figure 1 but not in the text). This should be elucidated in the statistical analysis paragraph
  • Which Statistics program was used to perform the analyses?
  • It would be interesting to add CRP values for the two groups
  • How complete were the data at the different timepoints of blood sampling?

Results:

  • The presented data refer only to the matched population, this should be clearly stated in Table headings and in the text
  • At least baseline characteristics of the unmatched groups should be presented
  • The authors state (P 6 lines 146 -148) “ However, […] a similar extent of reduction […] to about the same level [….]” showing no statistical significance. And P 6 lines 148 -150) “NLR reduction […] did not achieve statistical significance […]”. However, they state P 13 Lines 190 – 192: “The effect of the TAVI on the reduction of the NLR inflammatory marker in the reduced LV group caught up to the NLR of the preserved LV function group after 191 six-months follow-up“. And “Our findings are in agreement with the recent publication of Baratchi et al. [25], who 204 found an anti-inflammatory effect of TAVI reflected by a significant reduction of the NLR.” The Reviewer can not see the data supporting these findings. Please add or present it more clearly in the result section/figures/tables.
  • Page 12, lines 176, how is MACE defined? Abbreviation?
  • Table 5: which bleeding of VARC 2 definition life-threatening, major, minor, all? Same with Stroke and acute kidney injury.
  • Figure 4: Clinical follow-up of 10 years with almost no patients (see number at risk). Possibly data of shorter follow-up should be presented…
  • patient number is too small to allow for certainity of findings

Minor comments

  • Typos should be addressed
  • abbreviation NLR is spelled out more than once, further it is abbreviated a few times as NL ratio and other times as NLR. Please be consistent with one abbreviation

Author Response

Attached is a point by point letter addressing the comments and the changes made accordingly

Reviewer 2 Report

1. lines 62-63- please clarify what BEV and SEV represent in the manuscript

2. lines 94-96- why were patients with LVEF <25% excluded? Why not include them as a third group and compare their outcomes?

3. Figures 2 and table 2 as well as figure 3 and Table 3-  seem to convey the same information please condense the information 

4.Table 4- is extremely difficult to understand. Please simplify and condense the information and summarize in the results section

Author Response

(The authors gave the same response as above.)

Round 2

Reviewer 1 Report

The Authors addressed the raised issues.